# GLEE: A Unified Framework and Benchmark for Language-based Economic Environments

## Abstract

Large Language Models (LLMs) show significant potential in economic and strategic interactions, where communication via natural language is often prevalent. This raises key questions: Do LLMs behave rationally? How do they perform compared to humans? Do they tend to reach an efficient and fair outcome? What is the role of natural language in strategic interaction? How do characteristics of the economic environment influence these dynamics? These questions become crucial concerning the economic and societal implications of integrating LLM-based agents into real-world data-driven systems, such as online retail platforms and recommender systems. While the ML community has been exploring the potential of LLMs in such multi-agent setups, varying assumptions, design choices and evaluation criteria across studies make it difficult to draw robust and meaningful conclusions. To address this, we introduce a benchmark for standardizing research on two-player, sequential, language-based games. Inspired by the economic literature, we define three base families of games with consistent parameterization, degrees of freedom and economic measures to evaluate agents' performance (self-gain), as well as the game outcome (efficiency and fairness). We develop an open-source framework for interaction simulation and analysis, and utilize it to collect a dataset of LLM vs. LLM interactions across numerous game configurations and an additional dataset of human vs. LLM interactions. Through extensive experimentation, we demonstrate how our framework and dataset can be used to: (i) compare the behavior of LLM-based agents in various economic contexts; (ii) evaluate agents in both individual and collective performance measures; and (iii) quantify the effect of the economic characteristics of the environments on the behavior of agents. Our results suggest that the market parameters, as well as the choice of the LLMs, tend to have complex and interdependent effects on the economic outcome, which calls for careful design and analysis of the language-based economic ecosystem.

## 1 Introduction

Recent research has increasingly focused on the capabilities of Large Language Models (LLMs) in decision-making tasks, revealing their potential to operate as autonomous agents in various economic environments, that typically require involving complex strategic thinking Horton (2023); Wang et al. (2023); Zhu et al. (2024); Li et al. (2024); Sun et al. (2025). Applications of LLM-based agents in such environments include Chess-playing Feng et al. (2024), task-oriented dialogue handling Ulmer et al. (2024), financial advisement Lakkaraju et al. (2023) and beyond. The promising capabilities of LLMs in strategic decision-making call for the study of these agents' behavior through the lens of game theory, e.g. whether and when LLM-based agents naturally converge to Nash-equilibrium Guo et al. (2024), and how well they learn to cooperate in repeated interactions Akata et al. (2023). The rise of LLM agents also calls for a clear benchmark for assessing the extent to which such agents behave rationally Raman et al. (2024).

An important property of many real-world economic environments is that communication between agents typically occurs through *natural language*. While economic modeling usually abstracts away these nuances for the sake of keeping the model simple and tractable, in practical scenarios the fact that natural language is involved may significantly affect the interaction outcome. For instance, in bargaining between two parties, the same offer can be interpreted very differently depending on how it is framed. Consider the following two offers for a business partnership: **(a)** *"We propose a 40-60*

*split as we believe your expertise will drive the majority of the success."* **(b)** *"We propose a 40-60 split because we've already invested significant resources and need a lower share to maintain balance."* Both offer the same numerical terms, but the framing in each case may lead to different reactions based on how the rationale is presented. A similar rationale applies to a broader range of economic scenarios. To illustrate this further, let us consider the following representative use cases:

- **Bargaining.** Alice and Bob co-own a startup valued at a million dollar, and must decide how to divide the proceeds from its sale. To reach an agreement, they take turns proposing how to split the shared value, with each delay in reaching a decision reducing the overall value for both parties. The interaction often involves free language in the proposal exchange.
- **Negotiation.** Alice aims to sell a product to a potential buyer, Bob. They negotiate by taking turns proposing prices, with Alice deciding whether to accept Bob's offer and sell, and Bob deciding whether to accept Alice's price and buy, continuing this process until they reach an agreement or the negotiation concludes. The negotiation often involves natural language, such as descriptions of the product, which can influence the outcome.[1]
- **Persuasion.** Alice is a seller trying to sell a product to Bob at an exogenously set fixed price. Alice knows the true quality of the product, while Bob only has a rough expectation of the product's quality. Alice sends a message to persuade Bob to buy the product, aiming to convince him of its value, regardless of its true quality. Bob, however, benefits only if the product is of genuinely high quality. The interaction then repeats for multiple rounds, with the product quality in each round being independently drawn. In such a scenario, Alice balances between her maximizing one-shot gain and building positive reputation, and linguistic communication can play a significant role.

These three types of interactions are inspired by influential models in the economics literature. They are also broad and flexible enough to capture a wide variety of real-world applications. The bargaining model, inspired by the seminal work of Rubinstein (1982), forms the basis for understanding how parties negotiate the division of shared resources, applicable in scenarios such as business mergers, partnership dissolution, and legal settlements. The negotiation game reflects celebrated models of bilateral trade Myerson & Satterthwaite (1983); McAfee (1992) where buyers and sellers negotiate over the prices of an indivisible good, which often arises in various applications, including real estate transactions, corporate acquisitions, and e-commerce platforms. Lastly, the persuasion model draws on classical models of information asymmetry and strategic communication Akerlof (1978); Crawford & Sobel (1982); Farrell & Rabin (1996), which play a pivotal role in advertising, marketing, political campaigning, and recommendation systems.[2]

The AI and NLP communities have recognized the potential of integrating natural language into stylized economic models of bargaining, negotiation and persuasion. They also recognize the importance of studying the capabilities and limitations of LLM-based agents within such frameworks. Research increasingly focuses on evaluating and improving LLM agents in bargaining and negotiation settings Abdelnabi et al. (2023); Deng et al. (2024); Xia et al. (2024); Bianchi et al. (2024); Noh & Chang (2024); Hua et al. (2024). In the realm of persuasion games, recent studies have introduced language-based frameworks that explore the design of optimal information transmission policies Raifer et al. (2022) and develop methods for generating data to predict human players' behavior Apel et al. (2022); Shapira et al. (2025; 2024).

Although each of these economic setups has been studied in the context of LLM-based agents, the approaches have varied widely across different studies. Each paper often adopts distinct modeling and implementation assumptions, poses unique research questions, and applies diverse evaluation criteria, making it challenging to compare results and draw broader conclusions about the capabilities and limitations of LLM-based agents in economic environments. Moreover, the vast majority of the existing game-theoretic benchmarks for LLMs do not even include setups in which agents are allowed to communicate in natural language Duan et al. (2024); Huang et al. (2024); Guo et al. (2024); Wang et al. (2024).

---

[1]Notice that in the economic literature, the terms "bargaining" and "negotiation" are often used interchangeably to describe both the division of a divisible good and the process of price negotiation. In this paper, we use the terminology "bargaining" to describe the former and "negotiation" for the latter, for ease of exposition. We refer to Appendix C for further discussion on the differences between the two games.

[2]These three fundamental game families are *sequential games*, in which players act in turns, unlike in simultaneous games. We discuss the importance of studying such games in the context of language-based environments in Appendix B.

To enhance the real-world reliability of LLM-based agents, it is essential to establish a clear benchmark that provides a standardized framework for modeling these economic interactions. This benchmark would ensure comparability across studies and enable generalization of findings, facilitating a deeper understanding of how various factors influence interaction outcomes and leading to more robust and reliable conclusions about the performance of LLM-based agents in real-life economic situations.

**Our Contribution**   We introduce GLEE, a unified framework for **G**ames in **L**anguage-based **E**conomic **E**nvironments,[3] focusing on the case of two-player games.[4] Our framework provides a principled way to evaluate the performance of LLM agents and human players in a wide class of fundamental language-based economic scenarios. Central to the framework is a clear and comprehensive parameterization of the space of all bargaining, negotiation and persuasion games (as described above). It defines degrees of freedom and evaluation metrics that are consistent across economic contexts. The generality of the space of games spanned by our parameterization follows from the richness of the degrees of freedom considered, which include the game horizon (number of rounds within each game), information structure (whether agents know each others' preferences or not), and communication form (whether agents communicate via free language or structured messages). The framework is implemented as an open-source codebase that allows researchers to instantiate a wide range of economic games and evaluate LLM behavior within them. While the specific LLMs used in experiments may evolve or become outdated, the evaluation setup and metrics remain valid. A detailed comparison between GLEE and existing benchmarks is provided in Appendix A.

Using this framework, we have collected a dataset of LLM vs. LLM games, comprising 587K decisions made by LLM-based agents across more than 80K games, involving 13 different LLMs. The framework enables controlled experimentation across a wide range of language models, game types, and experimental conditions, making the dataset a valuable resource for evaluating LLM behavior in diverse economic settings. It also supports in-depth analyses of how the parameters of the economic environment affect agent strategies and shape interaction outcomes. Additionally, we developed an interactive interface that enables human participants to compete against LLM-based agents across various game configurations. Using this setup, we conducted experiments and compiled a complementary dataset of human vs. LLM interactions.

Through extensive analysis of the collected data, we uncover meaningful behavioral and economic phenomena. First, we find that the economic outcome, measured by efficiency, fairness, and agent self-gain, is significantly shaped by market parameters, including the information structure, communication form, and interaction horizon. Second, we show that there are no absolute best-performing models in terms of any of the evaluation measures, and that the performance of one LLM strongly depends on its competitor's choice of LLM. Finally, we compare LLMs to human participants and find that human behavior is markedly more extreme: in each setting, they either outperform all LLMs or fall behind them entirely, depending on the game and their assigned role. Together, these findings show how GLEE can be used not only as an infrastructure for research, but also as a tool for deriving concrete insights into economic reasoning and strategic behavior in language-based interactions.

## 2  GAME FAMILIES AND PARAMETRIZATION

In this section, we formally define the three game families discussed in the introduction: bargaining, negotiation and persuasion. In Appendix C we further discuss the related economic literature, and review some well-known theoretical results concerning these games. For each family, we provide a formal game-theoretic definition of the game (including the players, strategies and utilities), define degrees of freedom, and define the game outcome and evaluation metrics. While previous research has effectively tackled the challenge of evaluating the rationality of individual agents Raman et al. (2024), our focus extends to evaluating the outcome reached by the strategic behavior of the agents, using the fundamental economic notions of *efficiency* and *fairness*. [5]

---

[3]Code and data are available in `https://github.com/gleeframework/GLEE`.

[4]The two-player case is central, as it marks the shift from single-agent decision-making to strategic, communicative interaction. Such scenarios are common in real life, intuitive to analyze, and still capture the essential dynamics of more complex multi-agent settings.

[5]Metrics are normalized to $[0, 1]$, such that higher values indicate better (more efficient or fair) outcomes.

In our modeling, we focus on three main market characteristics that are critical for understanding the dynamics of LLM-based agents in economic interactions: *game horizon*, *information structure*, and *communication form*. The *game horizon* refers to the number of time periods during which the game is played and whether the length of the horizon is known or unknown to the agents.[6] This factor influences the strategies agents adopt, particularly in terms of long-term planning and anticipation of future moves. The *information structure* determines whether agents are aware of each other's preferences, impacting their ability to predict and respond to the actions of others. Lastly, the *communication form* specifies whether communication between agents occurs through free language or structured, concise messages, which affects the richness and clarity of the exchanges.

## 2.1 BARGAINING GAMES

The first family of games is inspired by the celebrated bargaining model of Rubinstein (1982). The model encompasses a class of bargaining games where two players, Alice and Bob, alternate offers over a time horizon $T$ (usually, $T = \infty$) to divide a fixed sum of money $M$ between them.[7] Importantly, in these games delays are costly, a concept captured by *discount factors* $\delta_A, \delta_B$ assigned to each player, reflecting the decreasing value of future payoffs as time progresses. Formally:

At each odd stage $t$, Alice offers a division $(p, 1 - p)$ for some $p \in [0, 1]$. Bob decides whether to accept or reject. If Bob accepts, the (Alice, Bob) utility vector is given by $M(\delta_A^{t-1} p, \delta_B^{t-1}(1 - p))$. At each even stage $t$, Bob offers a division $(q, 1 - q)$ for some $q \in [0, 1]$. Alice decides whether to accept or reject. If Alice accepts, the game terminates, and the (Alice, Bob) utility vector is given by $M(\delta_A^{t-1} q, \delta_B^{t-1}(1 - q))$. If no agreement is reached at any stage, the utilities are defined to be $(0, 0)$. In the standard version of the game, the time horizon is infinite and the discount factors are common knowledge (i.e., Alice and Bob know both $\delta_A$ and $\delta_B$).

In our experiments, we simulate a wide range of such bargaining games, differing in the following degrees of freedom: **(i)** whether or not the players know their opponent's discount factor (complete vs. incomplete information); **(ii)** whether or not the players know when the game terminates (finite vs. infinite); **(iii)** whether or not players communication involve natural language (structured vs. linguistic); **(iv)** the values of $\delta_A, \delta_B \in (0, 1)$ and $M$; and **(v)** the value of $T$ in the finite horizon case.

An outcome of the game is a pair $(t_{ev}, p_{ev})$, where $t_{ev}$ is the stage index at which the game terminated, and $p_{ev}$ is the share of $M$ that Alice obtained (without considering the discount in the utilities). When the game terminates without reaching an agreement, we define $t_{ev} = \infty$, and the gain for both players is zero. The evaluation metrics used to assess the economic outcome are *efficiency* and *fairness*. Efficiency is now measured by the normalized sum of Alice's and Bob's discounted payoffs at the time of agreement: $\delta_A^{t_{ev}-1} p_{ev} + \delta_B^{t_{ev}-1}(1 - p_{ev})$ if $t_{ev} < \infty$, and 0 if $t_{ev} = \infty$. Fairness is defined as the distance between the actual division and the fairest division, $1 - 4 \cdot (p_{ev} - \frac{1}{2})^2$ if $t_{ev} < \infty$, and 1 if $t_{ev} = \infty$.[8]

## 2.2 NEGOTIATION GAMES

In the second family of games, referred to as negotiation games, a seller (Alice) and a buyer (Bob) are negotiating over the price of a product. At the outset, Alice owns a product she subjectively values at $V_A$. The subjective valuation of the potential buyer, Bob, is $V_B$. To capture the notion of valuation scale in negotiation games, we parameterize $V_i = M \cdot F_i$ for $i \in \{A, B\}$, where $F_i \in (0, 1)$ is a factor parameter $M$ is a scale parameter.

As in the case of bargaining games, Alice and Bob alternate offers: at each odd stage, Alice posts a price and Bob decides whether to buy the product or move on to the next stage. At each even stage, Bob is the one to post a price and Alice decides whether to sell at this price or reject and move on to the next stage. The game is played for $T$ stages, which again can be either finite or "infinite" (i.e.,

---

[6]In economic theory, "infinite horizon" typically refers to a large, unspecified duration. Accordingly, we use the term "infinite horizon" to describe cases where the horizon is both large and unknown to the agents.

[7]It is clear that under a full rationality assumption, the amount of money does not play a role in the analysis. However, for human (or LLM) players, it is evident that the amount of money indeed matters.

[8]We consider the case of no trade as fair, since both players get the same utility. Obviously, this is also the least efficient outcome, which highlights the natural fairness-efficiency tradeoff.

large and unknown to both players). Unlike the bargaining game, here we assume no discount factors on the utilities, hence whenever a price $p$ is accepted, the utilities for Alice and Bob are $p - V_A$ and $V_B - p$ respectively. If no trade is made, then the utilities are defined to be $(0, 0)$.

In this class of games, we consider the following degrees of freedom: **(i)** whether or not the players know their opponent's product valuation (complete vs. incomplete information); **(ii)** whether or not the players know when the game terminates (finite vs. infinite); **(iii)** whether or not players communication involve natural language (structured vs. linguistic); **(iv)** the values of $F_A, F_B \in (0, 1)$ and $M$; and **(v)** the value of $T$ in the finite horizon case.

An outcome of a negotiation game is captured by $p_{ev}$, which is the price at which the product is sold when there is a trade, and defined to be $p_{ev} = \emptyset$ whenever no trade is made. We consider the following evaluation measures of the game outcome *fairness* and *efficiency*. When there is trade, fairness is measured by $1 - 4 \cdot \left( \frac{p_{ev} - p_f}{M} \right)^2$, where $p_f = \frac{V_A + V_B}{2}$ is the "fairest price".[9] When trade is not made, we define the fairness to be $1$ (i.e., maximal fairness) to reflect that no-trade does not change the default allocation of the product. Efficiency is defined to be $1$ in the following cases (and zero otherwise): **(a)** Alice values the product more than Bob, and does not sell it ($V_A \geq V_B$ and $p = \emptyset$); and **(b)** Alice values the product less Bob, and sells the product at a price that is beneficial for both players ($V_A \leq p \leq V_B$); When averaged over a large number of simulated games for a certain game configuration, this measure estimated the probability of the event "an efficient trade occurs when it should occur".

## 2.3 PERSUASION GAMES

In a persuasion game, a seller (Alice) tried to persuade a buyer (Bob) to buy a product at a fixed price $\pi$. Alice privately knows the true product quality, which can be either high or low. Bob only knows that the prior probability of the product being of high quality is $p$. Alice gets a utility of $1$ if Bob buys (regardless of the true product quality), and $0$ otherwise.

Bob values a high-quality product at $v > \pi$ and a low-quality product at $u < \pi$. Therefore, the utility of Bob from buying a high-quality product is $v - \pi > 0$, and from buying a low-quality product is $u - \pi < 0$. For simplicity, we normalize the price to $\pi = 1$ and the value of a low-quality product to $u = 0$. In addition, we consider a currency scale parameter $M$ that serves as a multiplicative term of Bob's utility function. Overall Bob gets a utility of $M(v - 1)$ from buying a high-quality product, $-M$ from buying a low-quality product, and $0$ from not buying the product.

The timing of a single round is as follows. First, Alice observes the product quality (which is realized to be high-quality w.p. $p$, independently of other rounds). Then, Alice sends a message to Bob. Lastly, Bob decides whether to buy or not to buy the product, and utilities are realized accordingly. The game then consists of $T$ such rounds. Alice's goal is to maximize her cumulative utility over time. We differentiate between two types of persuasion game setups: **(a) Long-living buyer.** The buyer is long-living, in the sense that he also aims to gain his cumulative utility over time. In this case, both players observe the entire interaction history; and **(b) Myopic buyers.** Buyers are myopic, in the sense the buyer of stage $t$ only cares about the utility obtained at stage $t$. In this case, each buyer observes the statistics of all previous rounds (i.e., % of rounds in which Bob bought the product, and % of rounds in which Bob bought a low-quality product).[10]

We consider the following degrees of freedom in persuasion games: **(i)** whether or not Alice knows Bob's high-quality product valuation $v$ (complete vs. incomplete information); **(ii)** whether or not the players know when the game terminates (finite vs. infinite); **(iii)** whether or not Alice's messages involve natural language (structured vs. linguistic); **(iv)** the values of $v$, $p$ and $M$; **(v)** the value of $T$ in the finite horizon case; and **(vi)** whether the game is with a long-living Bob or with myopic buyers.

---

[9]Notice that unlike all other metrics, the fairness metric in negotiation games is not normalized, as the price $p_{ev}$ is unbounded in principle. However, we observe that normalizing the difference $p_{ev} - p_f$ by the scale parameter $M$ results in a measure that is between 0 and 1 in 99.5% of the cases.

[10]That is, each buyer observes sufficient statistics from the entire history. This implementation detail is due to context length memory which is an inherent limitation of LLM agents, as well as to reducing cognitive load on human players.

Table 1: Parameters and their optional values used to define the 1,320 game configurations across bargaining, negotiation, and persuasion game families for data collection. $T = \infty$ means a very large value of $T$, unknown to the players. CI = Complete Information. MA = Textual messages allowed.

| | **Bargaining** | | **Negotiation** | | **Persuasion** |
|---|---|---|---|---|---|
| $\delta_A$ | 0.8, 0.9, 0.95, 1 | $F_A$ | 0.8, 1, 1.2, 1.5 | p | $\frac{1}{3}$, 0.5, 0.8 |
| $\delta_B$ | 0.8, 0.9, 0.95, 1 | $F_B$ | 0.8, 1, 1.2, 1.5 | v | 1.2, 1.25, 2, 3, 4 |
| $M$ | $10^2, 10^4, 10^6$ | $M$ | $10^2, 10^4, 10^6$ | M | $10^2, 10^4, 10^6$ |
| $T$ | 12, $\infty$ | $T$ | 1, 10, $\infty$ | T | 20 |
| CI | True, False | CI | True, False | CI | True, False |
| MA | True, False | MA | True, False | Messages type | Binary, Textual |
| | | | | Buyer type | Long-living, Myopic |
| In total | 384 configurations | In total | 576 configurations | In total | 360 configurations |

An outcome of the game is a tuple $(n_{ev}, k_{ev}, r_{ev})$, where $n_{ev}$ is the number of rounds in which the product was of high-quality, $k_{ev}$ is the number of rounds in which the product was of high-quality *and* the buyer bought the product, and $r_{ev}$ is the number of rounds in which the product was of low quality *and* the buyer did not buy the product. We define *efficiency* to by the proportion of rounds in which the product was sold out of all rounds in which the product was of high quality (i.e., $\frac{k_{ev}}{n_{ev}}$), and *fairness* to be the proportion of rounds in which the product was not sold out of all rounds in which the product was of low quality (i.e., $\frac{r_{ev}}{T - n_{ev}}$).

## 3 DATA COLLECTION AND STATISTICAL ANALYSIS

In this section, we describe the process of data collection and analysis. We developed a user-friendly game management system to facilitate data collection from the games described in §2. The system is written in Python, designed for ease of use and customization, enabling future researchers to seamlessly collect data. Integrating new language models is straightforward, allowing them to participate in any configurable setup. The interface allows users to effortlessly run data collection across multiple configurations while involving various LLMs in the process. Additionally, the system supports analyzing the collected data to gain insights into the performance and interaction patterns of different models. The system features a simple and intuitive interface, as illustrated in the screenshots provided in Appendix D.1. Prompt samples are described in Appendix D.2.

**Configurations** Since the game space defined by the parameters presented in §2 is infinite for each of the game families, it is clear that data cannot be collected from all possible games. Therefore, we attempted to cover the game space by selecting diverse values for each of the parameters defining the games, and we collected data from every possible combination generated by these parameters. Table 1 shows the parameters defining the groups from which we collected data. In total, we collected data from 1,320 different configurations: 384 configurations of bargaining, 576 configurations of negotiation, and 360 configurations of persuasion games.

**Data** For data collection, we utilized 13 state-of-the-art large LLMs, spanning multiple architectures and vendors. The full list of models is provided in Appendix D.3. The 169 possible pairs (including a language model playing against itself, with attention to the assignment for Alice or Bob) played across 1,320 different configurations. Each of the LLMs played as Alice in 2,500 bargaining games, 2,500 negotiation games, and 1,000 persuasion games, and an equal number of games as Bob. In total, we collected data from 80k games, with full statistics presented in Appendix D.4. To enable comparison with human behavior, we developed an interface for collecting data from games played between LLMs and humans, and gathered data from 3,405 human participants. Details about the human data collection process are provided in Appendix E.

**Statistical Analysis** To enable adequate comparisons between models that played in different game configurations, we employ a statistical framework that controls for variation in game structure and player composition. Since each model encountered only a subset of all possible game setups, raw metric values are not directly comparable. To address this, we fit linear regression models that predict each target metric based on the full parameterization of the game and the participating players. This allows us to estimate the independent contribution of each parameter while accounting

for confounding effects introduced by the game configuration. In doing so, we obtain normalized estimates of model behavior that can be compared across heterogeneous settings. We train a separate linear regression model for each combination of game family and evaluation metric (introduced in §2). The construction of the feature representation is detailed in Appendix F.1.

The estimated regression coefficients ($\beta$ values) enable us to quantify the impact of each parameter value on the metric relative to a predetermined default value. The list of default values is detailed in Appendix F.2. In addition to the magnitude of the effects, we also report 95% confidence intervals, computed using the standard procedure for linear regression coefficients (see e.g. Wooldridge (2016)). Appendix F.3 shows that the linear model performs comparably to state-of-the-art regression models on our prediction tasks, justifying its use in our analyses.

Importantly, metric values presented in this section are calculated by averaging over all possible game configurations in our dataset. These averages are therefore highly sensitive to the particular configurations and the configuration distributions in our dataset. To tailor the benchmark to specific applications, we recommend re-defining the parameter space and their distributions according to the economic context.[11]

## 4 ECONOMIC AND BEHAVIORAL INSIGHTS

In this section, we present the findings of our analysis, focusing on how different configurations and player compositions affect the outcomes of the games. We examine the general trends observed across all game families and identify specific characteristics that differentiate the performances of language models and human players. We structure our results around the following research questions: **Q1:** How do the market characteristics, such as information, game horizon, and linguistic communication, affect efficiency and fairness? **Q2:** How do different LLMs behave in strategic interactions, and which models achieve fair, efficient, and high self-gain outcomes? **Q3:** How do humans perform compared to LLMs?

Table 2: The effect of the market-defining parameters on the efficiency and the fairness of the game, measured in percentage point improvement relative to the naive parameterization. CI = Complete Information; MA = Textual Messages Allowed; MY=Myopic Buyers; Eff. = Efficiency, Fair. = Fairness; Conf. Int. = Confidence Interval. Bolded values indicate the highest metric values within the confidence interval, while underlined values indicate the lowest metric values.

(a) Bargaining

| CI | MA | T | Eff. | Fair. |
|---|---|---|---|---|
| – | ✓ | ∞ | **2.8** | **4.4** |
| – | – | ∞ | 2.2 | 2.8 |
| ✓ | ✓ | ∞ | 1.6 | 0.9 |
| ✓ | – | ∞ | | |
| – | ✓ | 12 | -0.2 | **4.4** |
| – | – | 12 | -0.8 | 3 |
| ✓ | ✓ | 12 | -1.8 | 0.9 |
| ✓ | – | 12 | -3.7 | -0.8 |
| Conf. Int. ∈ | | | ±0.5 | ±0.6 |

(b) Negotiation

| CI | MA | T | Eff. | Fair. |
|---|---|---|---|---|
| ✓ | – | ∞ | **17.4** | 0.2 |
| ✓ | ✓ | ∞ | **17.3** | 0.3 |
| ✓ | – | 10 | **16.7** | 0.1 |
| ✓ | ✓ | 10 | **16.3** | 0.1 |
| – | – | 10 | 9.2 | -1.2 |
| – | ✓ | 10 | 8.3 | -2.2 |
| ✓ | – | 1 | 6.8 | -0.9 |
| ✓ | ✓ | 1 | 5.5 | -0.3 |
| – | – | ∞ | 5.5 | -0.5 |
| – | ✓ | ∞ | 4.1 | -1.4 |
| – | – | 1 | | |
| – | ✓ | 1 | -2 | **0.2** |
| Conf. Int. ∈ | | | ±1.7 | ±0.3 |

(c) Persuasion

| CI | MA | MY | Eff. | Fair. |
|---|---|---|---|---|
| ✓ | – | – | **(max)** | **(max)** |
| – | – | – | -3.2 | -13.4 |
| ✓ | – | ✓ | -5.7 | **-0.7** |
| – | – | ✓ | -10.9 | -6.6 |
| – | ✓ | – | -12.2 | -15.6 |
| ✓ | ✓ | – | -13.6 | -12.7 |
| – | ✓ | ✓ | -22.4 | -22.2 |
| ✓ | ✓ | ✓ | -24.7 | -22.7 |
| Conf. Int. ∈ | | | ±2.5 | ±2.4 |

**Influence of Game Parameters (Q1)** Table 2 presents the effect of the market-defining parameters on fairness and efficiency, respectively, for each of the game families. From the table, we observe that market conditions have a decisive impact on the fairness and efficiency achieved in the game.

Across all game families, a prolonged interaction between Alice and Bob consistently enhanced both efficiency and fairness. However, the effects of complete information and textual message allowance

---

[11]For instance, one could ask whether agents' performance significantly differs in economic environments where inflation is high (translating into lower discount factors, in bargaining games). To evaluate such a scenario, one can simulate games in which the discount factor distribution fits these conditions, and re-evaluate agents' performance with respect to the new distribution of configurations.

Table 3: The effect of the Agent on the self gain, for each game family and role in the game.

| Family | Bargaining | | Negotiation | | Persuasion | |
|---|---|---|---|---|---|---|
| Model | Alice | Bob | Alice | Bob | Alice | Bob |
| human | **6.6 ± 1.1** | -17.1 ± 1 | -26.6 ± 1.1 | -14.5 ± 1.1 | **20.5 ± 4.1** | **54.8 ± 8.9** |
| llama-3.3-70b | 1.4 ± 0.7 | 0.7 ± 0.6 | 2.5 ± 0.6 | 4.2 ± 0.6 | 2.1 ± 2.4 | 11.2 ± 4.2 |
| claude-3-5-sonnet | 1.4 ± 0.7 | **2.6 ± 0.6** | 3 ± 0.6 | 4.5 ± 0.6 | 3.6 ± 2.4 | 12.1 ± 4.2 |
| claude-3-7-sonnet | 1.4 ± 0.7 | **2.1 ± 0.6** | 2.9 ± 0.6 | 4.4 ± 0.6 | 4 ± 2.4 | 11.3 ± 4.2 |
| gpt-4o | 1 ± 0.7 | 0.2 ± 0.6 | 2.8 ± 0.6 | 4.2 ± 0.6 | 4.3 ± 2.4 | 15.1 ± 4.2 |
| gemini-2.0-flash | 0.3 ± 0.7 | -0.1 ± 0.6 | **3.5 ± 0.6** | 3.7 ± 0.6 | -0.1 ± 2.4 | 21.2 ± 4.2 |
| gemini-1.5-flash | 0 ± 0 | 0 ± 0 | 0 ± 0 | 0 ± 0 | 0 ± 0 | 0 ± 0 |
| grok-2-1212 | -0.3 ± 0.7 | 0.5 ± 0.6 | 3 ± 0.7 | 4.4 ± 0.6 | 4.6 ± 2.4 | 11.2 ± 4.2 |
| gpt-4o-mini | -0.4 ± 0.7 | -1.5 ± 0.6 | 2.1 ± 0.6 | 3.5 ± 0.6 | 0.1 ± 2.4 | 10.1 ± 4.2 |
| llama-3.1-405b | -0.9 ± 0.7 | -0.2 ± 0.6 | **4 ± 0.6** | 4.7 ± 0.6 | -0.1 ± 2.4 | 21 ± 4.2 |
| mistral-large-2411 | -1 ± 0.7 | -1.3 ± 0.6 | 2.5 ± 0.6 | **5.3 ± 0.6** | 2.6 ± 2.4 | 28.3 ± 4.2 |
| gemini-1.5-pro | -1 ± 0.7 | -1.5 ± 0.6 | 2.4 ± 0.6 | **5 ± 0.6** | 6.3 ± 2.4 | 25.3 ± 4.2 |
| gemini-2.0-flash-lite | -1.4 ± 0.7 | -0.3 ± 0.6 | 2.9 ± 0.6 | 2.9 ± 0.6 | 1.1 ± 2.4 | 9.7 ± 4.2 |
| o3-mini | -6.9 ± 0.7 | 1.4 ± 0.6 | 2.4 ± 0.6 | 3.8 ± 0.6 | 3.4 ± 2.4 | 24.2 ± 4.2 |

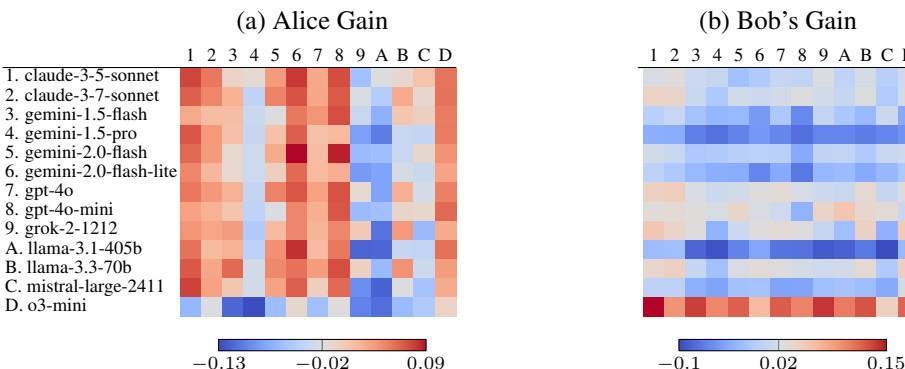

(a) Alice Gain      (b) Bob's Gain

1. claude-3-5-sonnet
2. claude-3-7-sonnet
3. gemini-1.5-flash
4. gemini-1.5-pro
5. gemini-2.0-flash
6. gemini-2.0-flash-lite
7. gpt-4o
8. gpt-4o-mini
9. grok-2-1212
A. llama-3.1-405b
B. llama-3.3-70b
C. mistral-large-2411
D. o3-mini

Figure 1: The effect of the identities of the two players (rows: Alice, columns: Bob) on self-gain in Bargaining games, reported relative to the mean outcome.

varied by game type. In bargaining games, complete information reduced both efficiency and fairness, while allowing messages improved both metrics. Conversely, in persuasion games, the pattern was nearly reversed: allowing messages degraded both efficiency and fairness. This held true for scenarios in which Bob was either a casual or a repeat buyer. In contrast, complete information improved fairness in both settings without significantly affecting efficiency. In negotiation games, complete information improved both metrics, whereas message allowance enhanced fairness without impacting efficiency.[12]

Interestingly, our results suggest that the effect of complete information on fairness and efficiency may depend on whether message allowance is enabled. In particular, in bargaining games, complete information appears to reduce efficiency and fairness only when linguistic messages are allowed. This interaction is noteworthy because traditional economic models typically abstract away from natural language communication. Our findings suggest that some classic theoretical insights, derived under structured or language-free assumptions, may not be applicable when rich linguistic communication is present. While a comprehensive theoretical investigation is out of scope, we believe this is a promising direction for future work at the intersection of language and economic theory.

**LLMs Performance (Q2)** Table 3 presents the agents' performance as Alice and as Bob in each of the game families in terms of self-gain. Nevertheless, the choice of which LLM to deploy in order to maximize utility is more nuanced than simply selecting the model with the best overall performance. Figure 1 presents the payoffs of Alice (left) and Bob (right) in bargaining games as a function of

---

[12]We hypothesize that the impact of free language communication differs across game types according to the presence of ground truth. In persuasion games, an objective truth (e.g., product quality) allows agents to build trust through consistent behavior, making free-form language less essential and potentially harmful if it introduces noise. In contrast, bargaining and negotiation games lack an objective ground truth, so language becomes essential for coordination, signaling intentions, and achieving mutually beneficial outcomes.

the identity of the models playing each role. Notably, when Bob is fixed to play with the LLM that yields the highest utility for him across the evaluated LLMs (*claude-3.5-sonnet*, see Table 3), Alice maximizes her payoff by selecting to play with *mistral-large-2411*. However, this model generally performs poorly when playing as Alice (ranked 11th out of 13). [13]

Figures 21, 22 and 23 in Appendix G.1 show how the pair of models playing negotiation games influenced the efficiency and fairness of the game. These tables suggest that there is no single model that maximizes both efficiency and fairness against all other models, and that compatibility between models plays an important role in shaping these metrics. Furthermore, across all game families, the scenario in which Alice and Bob employed the same LLM neither improved nor degraded the efficiency or fairness of the game, compared to the scenario in which they used different LLMs.

**Human Performance (Q3)**    Human performance deviates significantly from that of LLMs across the evaluated tasks, in terms of self-gain. In persuasion games, humans consistently outperformed all LLMs by a substantial margin. However, in negotiation games, their self-gain was the lowest among all other LLMs. An intriguing pattern emerged in the bargaining games: while humans performed considerably worse than language models when playing the role of Bob, they significantly outperformed the models when playing as Alice, despite the game's ostensibly symmetric structure in terms of information. A plausible explanation for this phenomenon lies in a well-documented behavioral bias known as the anchoring effect Tversky & Kahneman (1974), where the way information is presented can heavily influence human decision-making, even when the substantive content remains unchanged. In our setup, Alice initiates the interaction and anchors the bargaining by making the first offer, effectively setting the frame for the discussion. Unlike LLMs, humans tend to anchor their negotiation behavior to the initial proposal, often in a consistent yet irrational manner. For example, when Bob is a human and Alice is a LLM, the correlation between Alice's first offer and her final payoff was 0.63, indicating a strong anchoring effect on the human participant. Conversely, when Alice was human and Bob was a LLM, this correlation dropped to just 0.18, suggesting that language models are less influenced by the initial anchor and evaluate offers more rationally than humans, although still not in a fully rational manner.

## 5 CONCLUSION

We present GLEE, a framework for evaluating the behavior of Large Language Models (LLMs) in language-based economic games. The goal of GLEE is to provide a comparative tool for assessing the performance of LLMs in various economic scenarios and enable their comparison to human players. We defined the game space within three main families of games: bargaining, negotiation, and persuasion, and introduced metrics to measure player performance. We developed a framework that allows for large-scale data collection from games between diverse LLMs and created an interface that facilitates the collection of data from games involving human players and LLMs.

Through this interface, we gathered data from a variety of economic games, including both LLM vs. LLM and human vs. LLM interactions. Our analysis uncovered several notable phenomena: the varying impact of market parameters on economic outcomes, the interdependence between the game outcome and the type of LLMs used for both players, and the deviations of human performance from that of LLMs across different game families. We believe that GLEE offers a valuable foundation for advancing interdisciplinary research into the economic implications of LLMs, enabling systematic exploration of how AI agents and human decision-makers strategically interact and influence economic outcomes.

---

[13]One can think of a *meta-game* in which Alice and Bob (the users) choose LLMs to represent them in an economic game, aiming to maximize their expected utility (over all game parameter realizations). The strategy space consists of available LLMs, and players know only the game family, rather than its specific parameters. Under this setup, the only pure-strategy Nash equilibria are: **Bargaining**: Alice selects `claude-3-5-sonnet` and Bob selects `claude-3-7-sonnet`; **Negotiation**: Alice selects `gemini-2.0-flash` and Bob selects `gemini-1.5-pro`, or Alice selects `llama-3.1-405b` and Bob selects `grok-2-1212`; **Persuasion**: Alice selects `gemini-1.5-pro` and Bob selects `mistral-large-2411`.

ETHICS STATEMENT

This paper aims to provide a platform for experimenting with agents in language-based economic environments. Naturally, this line of research may have various societal and ethical implications, as we now discuss.

First, studying the economic aspects of LLM-based agents has the potential to enhance the ability of agent designers to control and optimize the behavior of these agents. This capability can be utilized for a variety of purposes, ranging from encouraging self-interested behavior at the expense of other participants in the environment (e.g., for maximizing revenue in competitive settings) to promoting efficient trade and fair behavior, or any combination of these sometimes non-aligned objectives. As this research increases the power of LLM-based agents in economic environments, it is essential to emphasize that with great power comes great responsibility. We call for the responsible and ethical use of these emerging capabilities to ensure they are leveraged for socially beneficial purposes rather than exploitative ones.

Furthermore, our framework demonstrates the capability of collecting data from human players to understand the differences and similarities between LLMs and humans in economic environments. While this line of research has the potential for a strong scientific contribution, particularly in the field of behavioral economics, it also raises several ethical considerations. The collection of human data must be conducted with careful regulation and adherence to clear ethical guidelines. The entire process of data collection from human participants is elaborated in Appendix E.

In addition to the challenges associated with the collection of human data, enhancing our understanding of LLMs from the lens of human behavior carries inherent risks. For instance, this research could enhance our ability to design LLM agents that are difficult to distinguish from real humans. Such capabilities could be misused for malicious purposes, including deception or manipulation. While the answer to whether these capabilities could be used for harmful causes is likely yes, we believe that the benefits of pursuing this line of research outweigh the risks when balanced with proper regulations. We advocate for pushing research forward while ensuring that any new technologies are accompanied by safeguards to prevent harmful usage, particularly when human-like LLMs are involved.

Our proposed framework can make these research directions more accessible to researchers from the ML community and beyond, thereby encouraging a broader understanding of LLM behavior in economic contexts. However, as such accessibility increases, it is crucial to maintain ethical oversight and foster an open dialogue on potential misuse. We encourage researchers to use our framework with full transparency and careful attention to potential misuse and negative consequences.

**LLM Usage Disclosure** In accordance with ICLR 2026 policy, we note that ChatGPT and Grammarly were used to aid or polish the writing of this paper. All conceptual contributions, experimental design, analysis, and conclusions are solely those of the authors.

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

## A    How GLEE Differs from Prior Benchmarks

To put in context the contribution of GLEE within the broader research landscape, we compare it to several recent benchmarks designed to evaluate the performance of LLMs in strategic settings. While these benchmarks differ in their choice of games, evaluation metrics, and interaction modalities, none focus on multi-turn economic games involving natural language communication. GLEE addresses this gap by offering a unified, richly parameterized framework for studying LLMs in sequential, language-based economic interactions. The following comparisons highlight where GLEE introduces new capabilities and how it complements or extends existing benchmarks.

**LLMs as Rational Players** Fan et al. (2023). Test the rationality of LLM in the light of game theory, using three simple one-shot games with no interaction between the players. GLEE supplies multi-round negotiation/bargaining/persuasion with free-text chat and direct human–agent comparison.

**GAMA-Bench** Huang et al. (2024). Introduce 8 single-turn games studied in game theory to test LLM rationality. The games are not economic games with little interaction between players. GLEE targets language–centric games, with games extensively studied in economic literature and extensive human trials.

**GTBench** Duan et al. (2024). Introduces a benchmark to test LLM rationality using board and card games. These games are not economic games, and natural language interaction does not happen. GLEE emphasis is on natural language *communication* (e.g., concessions, persuasion) within purely economic games.

**Game theoric LLM** Hua et al. (2024). Measures LLM behaviour on different multi-turn non-economic games extensively studied in game theory. The games they use have no extensive parameterization, and therefore few variants. GLEE is based on three economic games with heavy parameterization, allowing testing LLMs on hundreds of variants of the same game.

**TMGBench** Wang et al. (2024). Introduces an exhaustive benchmark with 144 $2 \times 2$ normal-form games, but these games are not economic games, are often synthetic, single-shot, and without communication between the players. While GLEE covers fewer games, it captures real dynamics with a multi-turn and natural language communication between the player.

**CompeteAI** Zhao et al. (2023). Introduce a one-competition world with multiple players; the metrics used are qualitative (LLM judgements). GLEE generalises across three economic game families with multiple parametrizations, using quantitative metrics derived from game theory research.

We summarize key differences in Table 4, highlighting where GLEE introduces new capabilities.

Table 4: Comparison of benchmarks evaluating LLMs in strategic games. GLEE stands out by focusing on multi-turn, natural language, economic interactions with extensive human evaluation and openness.

| Framework | Multi turns | NL Communication | Human Evaluation |
|---|---|---|---|
| **GLEE (ours)** | ✓ | ✓ | ✓ |
| Rational Players Fan et al. (2023) | ✗ | ✗ | ✗ |
| GAMA-Bench Huang et al. (2024) | ✗ | ✗ | ✓ |
| GTBench Duan et al. (2024) | ✓ | ✗ | ✗ |
| Game theoric LLM Hua et al. (2024) | ✓ | Structured | ✗ |
| TMGBench Wang et al. (2024) | ✗ | ✗ | ✗ |
| CompeteAI Zhao et al. (2023) | ✓ | Indirect | ✗ |

## B    Sequential vs. Simultaneous Games

A key feature of all these games is that they are sequential, meaning players take turns acting rather than acting simultaneously. This makes communication, and in particular language-based communication, a crucial element of the interaction. In sequential games, communication is typically direct, with players able to attach messages to their actions (e.g., in the bargaining example, the textual message is attached to the proposal). In contrast, in simultaneous games, communication

tends to be indirect, as players act independently and typically learn to cooperate by observing and reacting to past actions rather than through direct message exchange. A prime example of a simultaneous, language-based economic interaction is the competition among web publishers in search engines. Publishers simultaneously create content (usually in the form of textual documents) for their websites, and then gradually learn and adapt based on outcomes, such as their relative search engine rankings and exposure. However, they usually lack the opportunity for real-time communication or coordination during this process. These simultaneous, language-based interactions are well-studied within the information retrieval community, and the induced publishers' game is indeed modeled as a simultaneous game (see e.g. Raifer et al. (2017); Ben-Porat et al. (2019); Kurland & Tennenholtz (2022); Hron et al. (2022); Madmon et al. (2023; 2024); Nachimovsky et al. (2024), in which SEO competitions are modeled as simultaneous games). Our work is complementary, focusing on sequential settings where direct, language-based communication is prevalent.

## C  GAME FAMILIES THROUGH THE LENS OF ECONOMIC LITERATURE

In this section, we discuss the economic literature related to the three game families considered in this paper, and review some known theoretical results.

**Bargaining**  As mentioned in §2, the standard bargaining model of Rubinstein (1982) consists of two players, Alice and Bob, engaging in alternating offers for a finite horizon ($T = \infty$) with commonly known discount factors $\delta_A, \delta_B$. Our parameterization considers several additional degrees of freedom, such as finite vs. infinite time horizon, complete vs. incomplete information, and free language messages vs. structured and concise messages. In the standard model, Rubinstein (1982) showed that in the unique subgame-perfect equilibrium an agreement is reached in the first stage (i.e., utilities are not discounted), and the share of Alice is given by $Mp^*$, where $p^* = \frac{1-\delta_B}{1-\delta_A\delta_B}$.[14] The case of a finite horizon can be solved using backward induction, and typically results in a different outcome compared to the infinite case. As $T$ grows, the equilibrium outcome approaches the one of the infinite case. Extensions to incomplete information regarding the opponent's discount factor are typically more challenging, and some of them are studied in the literature, e.g. Rubinstein (1985).

**Negotiation**  In a negotiation game, Alice and Bob negotiate over the price of an indivisible good. The negotiation game differs from the bargaining games in several key aspects:

1. Negotiation involves an indivisible product (e.g., Alice's product), while bargaining focuses on dividing a divisible resource, such as money.

2. In negotiation, Alice and Bob may have different subjective valuations of the product, whereas in bargaining, both parties value the divisible resource similarly.

3. Negotiation has no discounting, so the utility remains constant over time. In bargaining, delays reduce the total value, encouraging faster agreement.

If the seller is uncertain regarding the buyer's valuation but has a prior belief distribution, a classical result by Harris & Raviv (1981) and Riley & Zeckhauser (1983) states that it is always optimal to sell the product at a fixed price. In contrast, if the seller does not have a prior belief over the buyer's valuation, and instead aims to minimize regret, then an optional pricing policy will be to randomly choose a price from a carefully chosen distribution Bergemann & Schlag (2011).

**Persuasion**  Our persuasion game follows the structure of a cheap talk game (Crawford & Sobel, 1982; Farrell & Rabin, 1996), where the sender (Alice) cannot commit to a signaling policy in advance, unlike in Bayesian persuasion models (Kamenica & Gentzkow, 2011). Under the particular payoff structure considered in our persuasion game, it is well-known that the cheap-talk game only admits a *babbling equilibrium*, i.e., an equilibrium in which all information is kept hidden (this is due to the strong misalignment of interests between the two players). In contrast, if the seller can *commit* to a signaling policy at the outset, as in Bayesian persuasion, then there exists a subgame-perfect equilibrium in which the seller commits to the following policy: When the product is of

---

[14]A subgame-perfect equilibrium is a strategy profile in which every player responds optimally in every hypothetical subgame of the game, including off-path scenarios that are not reached in practice. This solution concept can be seen as capturing a higher level of rationality compared to the alternative of Nash equilibrium.

high quality, the seller recommends buying the product with probability 1. When the product is of low quality, then the seller recommends with probability $q = \min\{\frac{p}{1-p}(v-1), 1\}$. This policy is also incentive-compatible, in the sense that the buyer always buys the product when the seller recommends buying.[15] While the long-living buyer case is well-studied in the economic literature, such games often admit multiple equilibria, which makes the games difficult to analyze and predict (Kim, 1996; Aumann & Hart, 2003). As for the case of myopic buyers, Best & Quigley (2024) draws a connection between the repeated cheap talk game and the case of one-shot Bayesian persuasion, leading to an elegant analytical solution of the repeated game. Intuitively, the repetitive nature of the game induces a reputation effect, which plays a similar role to the commitment power in standard one-shot Bayesian persuasion.

# D  LLM DATA COLLECTION

## D.1  GAME MANAGEMENT SYSTEM INTERFACE

In this appendix, we present the main features of our game management system through a series of screenshots. The system facilitates data collection and analysis from the games described in §2, offering a user-friendly and customizable interface.

The main interface of the system (Figure 2) provides three primary options: starting a new data collection, resuming a previously interrupted collection, and analyzing results from past experiments. When initiating a new data collection, users first select the game family and the participating LLMs (Figure 3), followed by defining the set of configurations to be executed (Figure 4).

Once the data collection is complete, users can move to the analysis phase. The data analysis module starts with selecting the data to be analyzed (Figure 5). After selecting the relevant data, users can use the parameter impact viewer (Figure 6) to visualize how game parameters influence various metrics (see §3). Additionally, the system provides a statistics table (Figure 7) that summarizes key characteristics of the collected data.

The system does not require any specialized hardware: data collection and analysis can be performed efficiently using a single CPU.

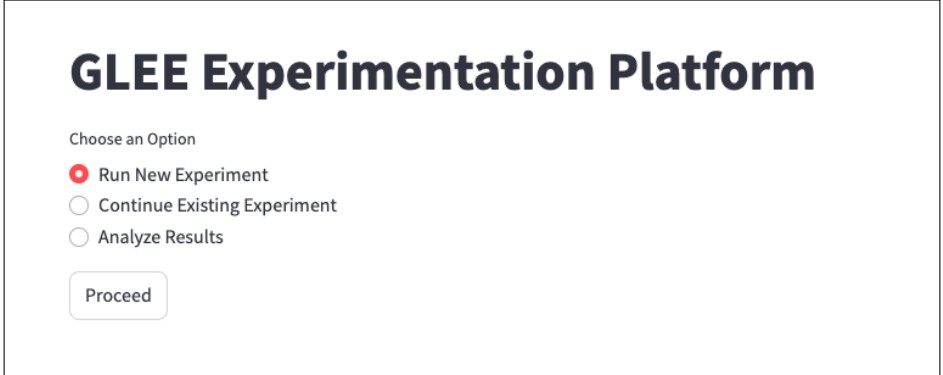

Figure 2: Main interface: Start a new data collection, resume a previous one, or analyze results from past experiments.

---

[15]In fact, the probability of lying $q$ is determined such that the buyer is indifferent upon receiving a recommendation, taking into account his belief updating, which relies on using Bayes's law and knowing the seller's committed policy.

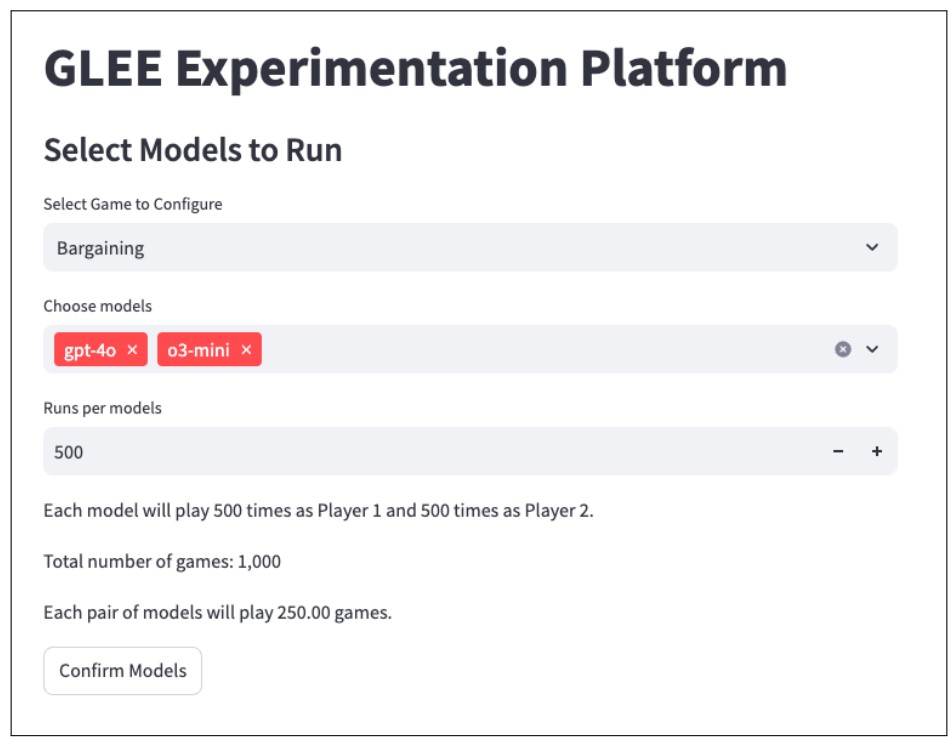

Figure 3: New data collection: Selecting game family and LLMs.

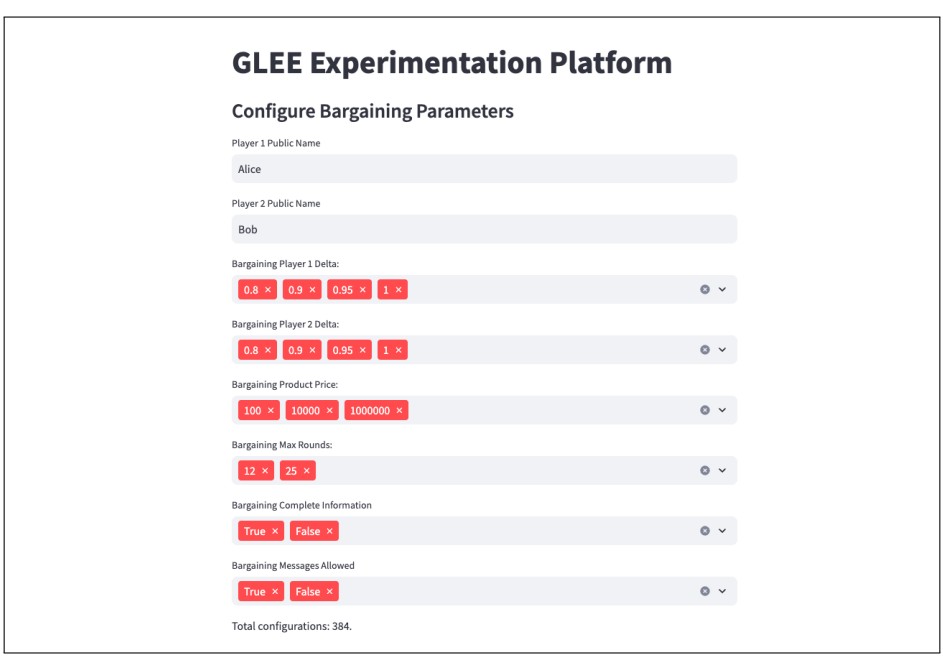

Figure 4: New data collection: Setting up configurations.

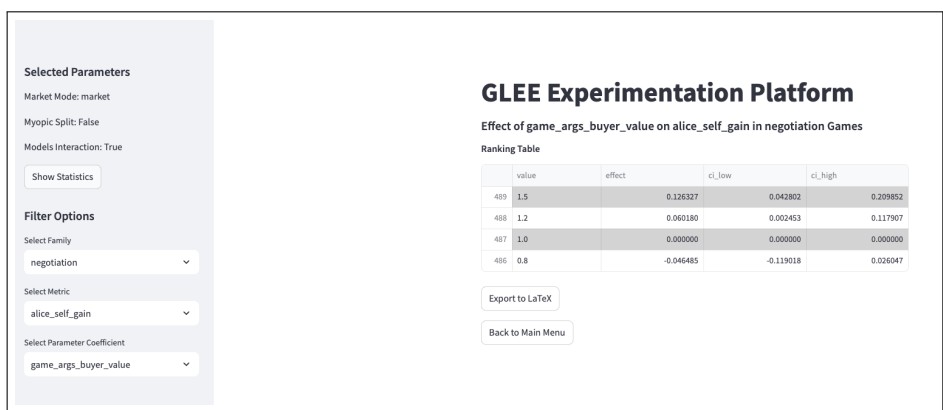

Figure 5: Data analysis: Selecting datasets for analysis.

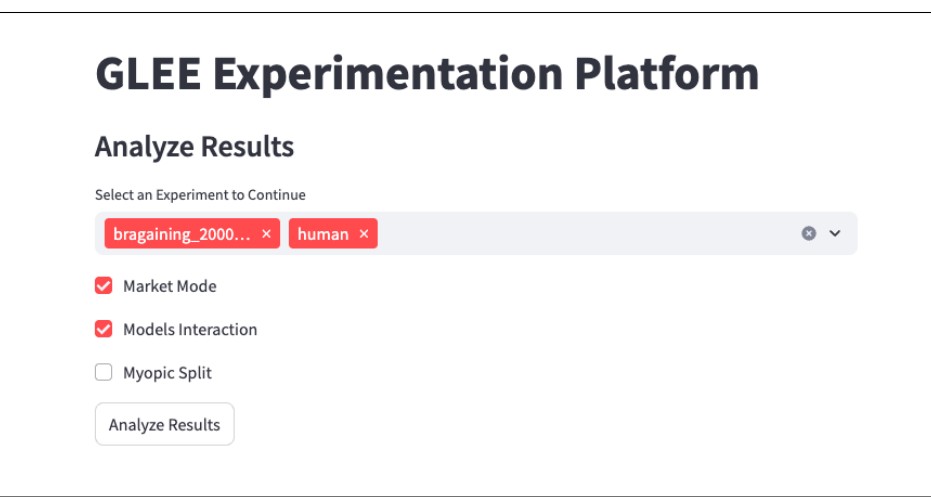

Figure 6: Data analysis: Visualizing the impact of game parameters on various metrics.

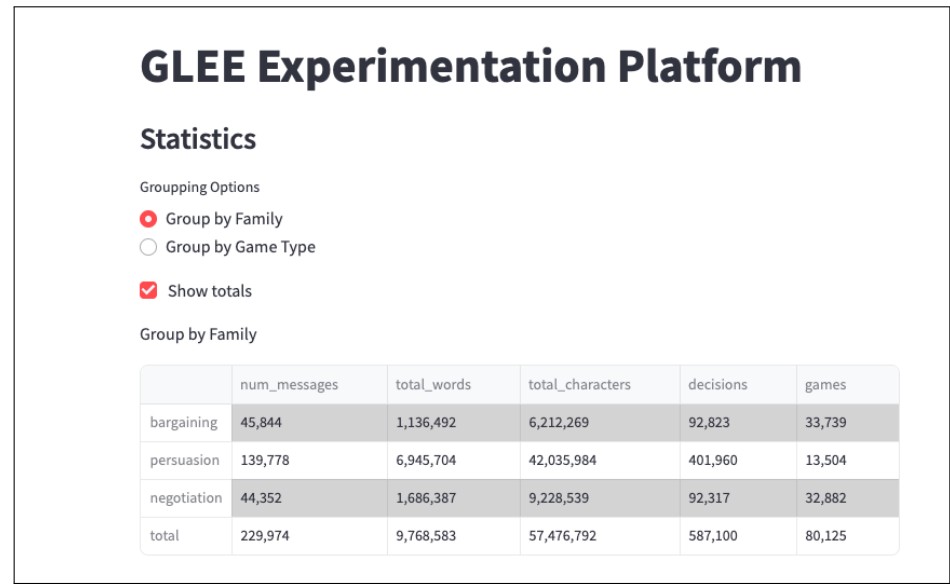

Figure 7: Data analysis: Statistics table summarizing key characteristics of the collected data.

## D.2 CONVERSATION STRUCTURES

In this appendix, we present an example of the conversation structures with the language models used for data collection in *GLEE*. The example is from a bargaining game, but the system's operating principle is the same in negotiation and persuasion games as well.

The prompts are divided into two types: (1) instructions and guidelines sent to the language model as a system message, and (2) game management messages sent to the language model as a user message. The language model performed its action, which was documented in the prompt as an assistant message. The system message begins with precise game instructions, which are constructed at the start of the game using the parameters that define the configuration. This is followed by a guideline specifying the format in which the LLM is required to respond. Each time the language model is required to make a decision (of any type), the language model receives a message containing all the information that has accumulated since its previous turn. This information appears as a system message and sometimes includes text written by the language model that played the second player.

It was possible to alter the prompt that introduces the game, explore different variations, and allow the language models to play different personas. While this could impact the behavior of the language models, budget limitations would have required us to collect less data from each configuration. Therefore, we chose to keep the prompt fixed rather than further defining the language model's setup. However, the system allows easy prompt customization to support future research.

Throughout the appendix, the system prompt (or parts of it) that Alice received are highlighted in red. Game management messages Alice received are shown in purple, and her response is shown in magenta. The system prompt that Bob received is highlighted in blue. Game management messages Bob received are shown in teal, and his response is shown in cyan.

---

Instructions to Alice:

Let's play a game. In this game, there are two players. You are playing as Alice. You and Bob need to divide 1,000$ between you. In each round of the game, one of the players will make a proposal for division and can describe their proposal in a few words. The other player will decide whether to accept the offer and finish the game, or to proceed to the next round, in which they will be the one making the proposal. We will relay messages between you and Bob. Beware of inflation! With each passing round, the money is worth 0% less for you. For Bob, the money is worth 10% less. You have 10 rounds to divide the money, or both of you will get nothing!

Instructions to Bob:

Let's play a game. In this game, there are two players. You are playing as Bob. You and Alice need to divide 1,000$ between you. In each round of the game, one of the players will make a proposal for division and can describe their proposal in a few words. The other player will decide whether to accept the offer and finish the game, or to proceed to the next round, in which they will be the one making the proposal. We will relay messages between you and Alice. Beware of inflation! With each passing round, the money is worth 10% less for you. For Alice, the money is worth 0% less. You have 10 rounds to divide the money, or both of you will get nothing!

---

Alice turn.

Alice got this message from the system:

'Round 1
Send your offer to divide $1,000 and a message to Bob.'

Alice guideline:

Send your offer to divide 1,000$ and the message you attached in the JSON format:
{"alice_gain": The part that you will receive in your offer,
"bob_gain": The part that Bob will receive in your offer,
"message": The message you pass to Bob}

Alice's response:

```json {"alice_gain": 900, "bob_gain": 100,
"message": "Let's start fair. I'll take the bigger share, but you get something too."} ```

---

Bob turn.

Bob got this message from the system:

Round 1
Alice's offer:
# Alice's message: Let's start fair. I'll take the bigger share, but you get something too.
# Bob gain: 100
# Alice gain: 900
Do you accept this offer?

Bob guideline:

Answer with {"decision": "accept"} or {"decision": "reject"}

Bob's response:

```json {"decision": "reject"} ```

---

Bob turn.

Bob got this message from the system:

You have chosen to reject Alice's offer from round 1.
Round 2
Send your offer to divide $1,000 and a message to Alice.

Bob guideline:

Send your offer to divide 1,000$ and the message you attached in the JSON format:
{"bob_gain": The part that you will receive in your offer,
"alice_gain": The part that Alice will receive in your offer,
"message": The message you pass to Alice}

Bob's response:

```
"'json {"bob_gain": 500,
"alice_gain": 500,
"message": "Let's split it evenly. It's the fairest way to start."} "'
```

---

Alice turn.

Alice got this message from the system:

Bob rejected your offer from round 1.

Round 2
Due to inflation, the money Bob gains is worth 10% less than in the first round. The money you gains is worth the same as in the first round.
Bob's offer:
# Bob's message: Let's split it evenly. It's the fairest way to start.
# Alice gain: 500
# Bob gain: 500
Do you accept this offer?

Alice guideline:

Answer with {"decision": "accept"} or {"decision": "reject"}

Alice's response:

```
"'json {"decision": "accept"} "'
```

The game is over.

### D.3  LLM LIST

Table 5 presents the 13 LLMs used for data collection (see 3), organized by model name, developer, and release year.

Table 5: Large Language Models used for data collection.

| Developer | Model Name | Release Year |
|---|---|---|
| Anthropic | Claude 3.5 Sonnet | 2024 |
| | Claude 3.7 Sonnet | 2024 |
| Google DeepMind | Gemini 1.5 Flash | 2024 |
| | Gemini 1.5 Pro | 2024 |
| | Gemini 2.0 Flash | 2025 |
| | Gemini 2.0 Flash Lite | 2025 |
| OpenAI | GPT-4o | 2024 |
| | GPT-4o Mini | 2024 |
| | GPT-O3 Mini | 2024 |
| xAI | Grok 2 | 2024 |
| Meta | LLaMA 3.1–40.5B | 2024 |
| | LLaMA 3.3–70B | 2024 |
| Mistral AI | Mistral Large | 2024 |

### D.4  STATISTICS

Table 6 contains statistics on the data we collected, presented in §3, which constitutes a contribution of the paper.

## E  HUMAN DATA COLLECTION

This appendix provides information on the human data collection interface described in §3, which facilitates data collection from GLEE games played between humans and LLMs.

Table 6: Statistics of data collected by family.

| Game Type | Games | Decisions | Messages | Words |
|---|---|---|---|---|
| Bargaining | 33.7K | 92.8K | 45.8K | 1.14M |
| Persuasion | 13.5K | 402K | 140K | 6.95M |
| Negotiation | 32.9K | 92.3K | 44.4K | 1.69M |
| Total | 80.1K | 587K | 230K | 9.77M |

One of the main objectives of collecting data from language games is to compare the behavior of LLMs with human behavior in economic and strategic situations. To facilitate this comparison, we developed an interface that allows human players to play all the language games that can be defined using GLEE. The interface transforms the various prompts presented to LLMs into user-friendly screens for human participants, displaying the prompt and requesting them to send messages and make decisions.[16] Through this interface, we enable human players to take on the role of one of the players while the other player is a pre-selected LLM. The interface is one of the contributions of this paper. Screenshots of the interface can be found in Appendix E.1.

The interface, developed using oTree (Chen et al. 2016), enables integration with Amazon's crowd-sourcing platform, mTurk,[17] through which we recruited 3,405 players who participated in various configurations against Gemini-1.5-flash. We chose Gemini-1.5-flash since it demonstrated strong performance compared to other LLMs and allowed comprehensive data collection due to its low usage cost. Since we aimed to collect multiple games from each configuration, we had to select a limited set of configurations for human participants to play. The process of selecting these sets is described in Appendix E.2. We collected human data from 195 different configurations: 78 of which were bargaining games, 60 of which were negotiation games, and 57 were persuasion games.

Each human player was allowed to play one game every 12 hours from each family of games. Human players were paid a base rate calculated at $6 per hour, plus an average bonus of $6 per hour. In total, we paid $2,245 to all players for their participation in the games. Bonuses were dependent on the configuration the human players played and their success in the game. The average bonus was known to players at the start of the game, aiming to encourage serious gameplay. To ensure players remained focused and made thoughtful decisions, we conducted two attention checks during the experiment, detailed in Appendix E.3. Players who failed in one of the attention checks were excluded from the dataset. To reflect the real-world significance of the magnitude of product prices (the parameter M) in each family of games, we defined the bonus as dependent on M: in configurations where $M = 10^2$, the average bonus was $3 per hour; where $M = 10^4$, the average bonus was $6 per hour; and where $M = 10^6$, the average bonus was $9 per hour.

### E.1 Screenshots of the Data Collection Interface

**General Application Structure** The structure of the application and the games consists of fixed parts and parts that vary between different game families. Each game starts with a screen where the player enters their name, followed by an instruction screen (Figure 8 in bargaining, Figure 12 in negotiation, and Figure 16 in persuasion). The instructions themselves can change depending on the type of game and the parameters of the game. In the middle of the game's rules on the instruction screen, there is a hidden prompt directing participants to enter a code word in the text box below. This test is designed to filter out unfocused participants. If the player fails this test, they are directed to a screen informing them of their failure and will not participate in the game. Otherwise, the game begins. In each round, both the human player and the LLM player perform an action, with the order depending on the game and configuration. An action could involve sending an offer to the other player (Figure 9 in bargaining games, Figure 13 in negotiation games and Figure 17 in Persuasion games) or responding to the other player's offer (for instance, Figure 10 in bargaining games, Figure 14 in negotiation games and Figure 18 in persuasion games). After both players have completed their actions, the human player is taken to a response screen (Figure 11), where he sees the decision of

---

[16]Since human players are accustomed to being addressed by their first name (rather than as Alice or Bob), we asked them to enter their name at the beginning of the game and referred to them by their name throughout the game. The player's name was the only difference between the human player and the LLM-based player.

[17]https://www.mturk.com/.

the LLM player. Afterward, if the game is still not over, the human player continues for another round. Once the game is finished, the player is taken to a quiz screen where they must answer a question related to the technical details of the game (Figure 15). If the human answer correctly, they are directed to the final screen (Figure 19), where they receive a code to enter on the mTurk website. If the player fails the final quiz, they do not receive a completion code and are taken to a screen that informs them of their failure in the quiz. In this case, we erase the game from our database.

### E.1.1 BARGAINING GAMES SCREENSHOTS

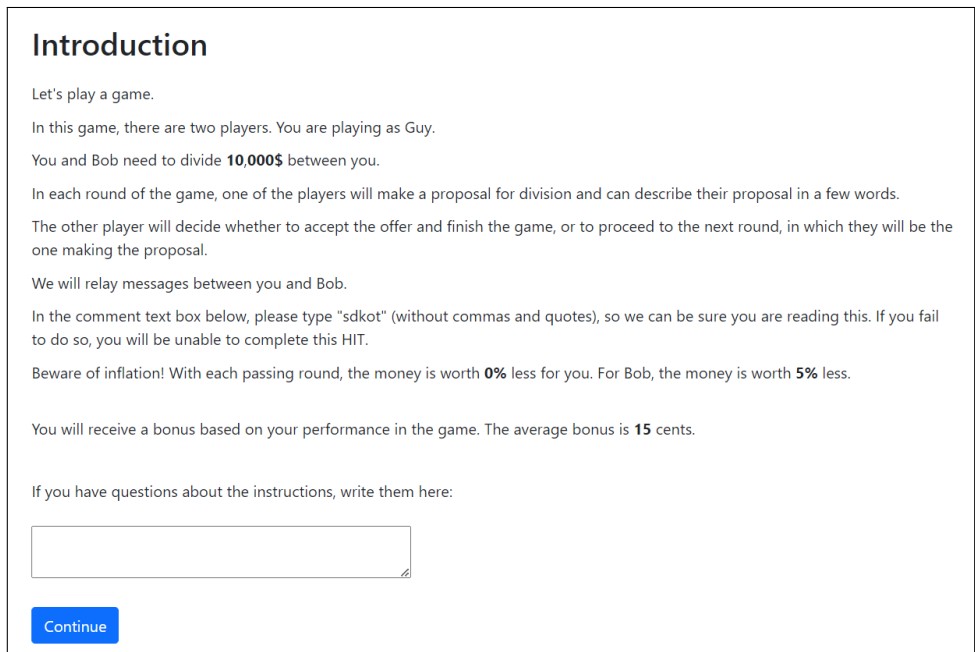

Figure 8: An example of an instruction screen shown to a human player at the start of a bargaining game.

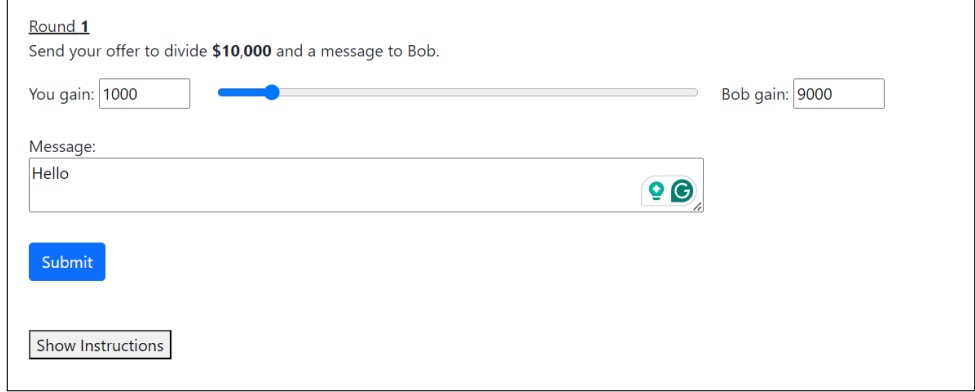

Figure 9: An example of a proposition screen shown to a human player during his first turn in a bargaining game.

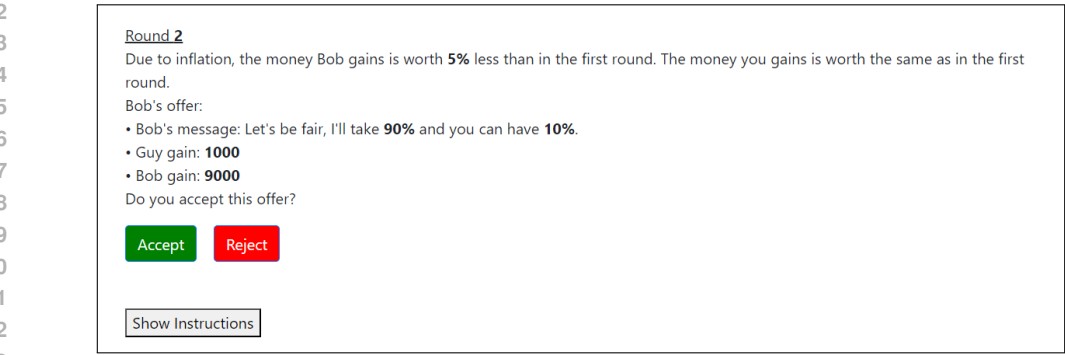

Figure 10: An example of a decision screen shown to a human player during his second turn in a Bargaining game.

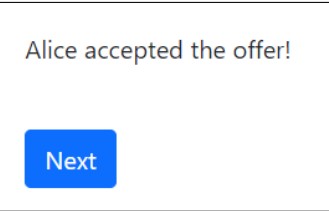

Figure 11: An example of a response screen shown to human players during his second turn in a Bargaining game.

### E.1.2 NEGOTIATION GAMES SCREENSHOTS

**Introduction**

You are Alice. You are selling one product that is worth no less then **$8,000** to you.

Bob is a potential buyer to whom you are offering the product. You don't know the value of the product to Bob.

You will offer Bob to buy the product at a price of your choice. Bob can either accept or reject the offer.

Your goal is to earn as much money as you can for the product.

If Bob rejects the offer, he can make a counteroffer to buy your product. You can either accept or reject his counteroffer. If you reject Bob's counteroffer, you can make a new counteroffer, and so on.

In the comment text box below, please type "sdkot" (without commas and quotes), so we can be sure you are reading this. If you fail to do so, you will be unable to complete this HIT.

You have **10** rounds to close the deal. However, at any moment, you can choose to stop the negotiation with Bob and sell the product to another buyer, John, who is willing to buy the product from you for **$8,000**.

You will receive a bonus based on your performance in the game. The average bonus is **15** cents.

If you have questions about the instructions, write them here:

Continue

Figure 12: An example of an instruction screen shown to a human sellers at the start of a Negotiation game.

Round **1**

Send your offer to Bob.

Offer: [ ]

Submit

Show Instructions

Figure 13: An example of a proposition screen shown to human sellers during his first turn in a Negotiation game.

Alice's offer: The product price will be **$10,000**.
You have three options:
(**1**) Accept Alice's offer, and buy the product for **$10,000**
(**2**) Reject Alice's offer and proceed to the next round, where you will send Alice a counteroffer
(**3**) Buy the product from Jhon for **$10,000**

Accept Offer     Reject Offer     Buy From Jhon

Show Instructions

Figure 14: An example of a decision screen shown to human buyers during his first turn in a Negotiation game.

# Final Quiz

## What is the minimum worth of the product for you in this game?

○ 4,000
○ 2,000
○ 32,000
○ 16,000
○ 8,000

Submit

Figure 15: An example of a final quiz screen shown to human sellers at the end of a Negotiation game.

### E.1.3 PERSUASION GAMES SCREENSHOTS

**Introduction**

You are Alice, a seller trying to sell products to Bob for **$100** each. We will relay your messages to Bob. It is known that some of the products are high-quality and some are low-quality.

The probability that a product is high-quality is **33.33%**, and Bob knows this.

A high-quality product is worth **$120** to Bob, and a low-quality product is worth only **$0** to them.

For **20** rounds, you will offer products to Bob, the buyer. At the beginning of each round, you will discover whether the product is high-quality or low-quality. Bob does not know this detail. You can send any message you wish to Bob, and based only on this message, they will decide whether to buy the product or not. At the end of the round, Bob will discover the quality of the product.

In the comment text box below, please type "sdkot" (without commas and quotes), so we can be sure you are reading this. If you fail to do so, you will be unable to complete this HIT.

The price is final, no discounts are allowed, and the product is non-refundable after purchase.

You will receive a bonus based on your performance in the game. The average bonus is **5** cents.

If you have questions about the instructions, write them here:

Continue

Figure 16: An example of an instruction screen shown to human sellers at the start of a Persuasion game.

**Round 1**
This round's product is low-quality.
Send Bob a message to help them decide whether to buy the product.

Message:

Submit

Show Instructions

Figure 17: An example of a proposition screen shown to human sellers during his first turn in a Persuasion game.

**Round 1**
Alice's message: Hi Bob, I have a great product for you today! It's priced at **$100**, a fantastic deal for its quality. Are you interested?

Would you like to buy the current product?

Accept    Reject

Show Instructions

Figure 18: An example of a decision screen shown to human buyers during his first turn in a Persuasion game.

Figure 19: An example of a game over screen shown to human players at the end of a Persuasion game.

### E.2 Selection of Configurations for Human Data Collection

In this appendix, we describe the method used to select which configurations human players would play and how many times each configuration would be played. Each configuration is defined by both the game parameters and the role of the human player (Alice or Bob).

For each family of games, we arbitrarily defined one configuration, which we referred to as the *main configuration*. This configuration contains the parameters that we deemed most interesting. For every main configuration, we collected data for both possible roles of the human player (Alice or Bob). In persuasion games, we defined two main configurations: one for recurring buyers and one for manipulated buyers, due to the significant theoretical differences arising from this parameter. We collected the largest amount of data from the main configurations to allow for more in-depth follow-up research.

Configurations that were identical to one of the main configurations except for one parameter[18] were called *variants of the main configuration*. We collected data for all of these configurations as well.

Additionally, we randomly sampled 5% of the other configurations and collected data from them as well. These configurations were referred to as *random configurations*.

Due to the desire to allocate the data collection budget to complex games, we did not collect any data from games in which the human player was required to play at most one round (single-round bargaining games and persuasion games in which the human player is a manipulated buyer).

For each category, we determined the number of games we wanted to collect from each configuration belonging to it. This decision was made based on budgetary considerations. In persuasion games, we were able to collect fewer games from each configuration due to the fact that these games take longer to complete (and therefore, the payment players received for participating in them was higher). Table 7 describes the number of configurations that belonged to each category for each game and the minimum number of games we collected from each category.[19]

Table 7: The number of configurations belonging to each category for each game family, as well as the number of human players who played each configuration within each category.

| Type | Bargaining | | Negotiation | | Persuasion | |
|---|---|---|---|---|---|---|
| | # config. | # games | # config. | # games | # config. | # games |
| main | 2 | 50 | 2 | 50 | 3 | 30 |
| variant | 40 | 25 | 22 | 25 | 30 | 8 |
| random | 36 | 15 | 36 | 15 | 24 | 5 |

---

[18] In bargaining games, we defined a change in both players' discount factors as a change in one parameter

[19] Since the games were played in parallel, for some configurations we collected more games than required. For 113 configurations, we collected one more game than required; for 4 configurations, we collected 2 more games than required; and for one configuration, we collected 3 more games than required.

### E.3 ATTENTION CHECKS FOR HUMAN PLAYERS

In this appendix, we describe the two attention tests that human players were required to complete. The purpose of these tests was to ensure that the human players stayed focused on the game and made conscious decisions, rather than random choices to finish the game as quickly as possible. The players were aware that their attention would be tested during the game, and they knew that they would not be paid for the task if they failed these tests. Out of the 4,652 players who started the game, 1,247 players (representing 26.8% of those who began the game) failed one of the tests and were not included in the final dataset.

The first test appeared on the instruction screen. Toward the end of the instructions, a line requested players to write the code word "sdkot" in a text box that appeared at the end of the instructions phase. Players who did not write this word were immediately disqualified and did not start the game, as they did not carefully read the instructions. A total of 412 players, representing 8.9% of those who began the game, failed this test.

The second test appeared at the end of the game. The human players were asked a basic question that depended on the family of the game they played. They were required to select the correct answer from four possible options. Players who participated in a bargaining game were asked about the inflation rate in their game (499 players, representing 22.7% of respondents, failed this question and were excluded from the dataset). Players who participated in a negotiation game were asked about the value of the product for them (68 players, representing 12.3% of respondents, failed this question and were excluded from the dataset). Players who participated in a persuasion game were asked about the price of products in the game (268 players, representing 18% of respondents, failed this question and were excluded from the dataset). In total, 835 players, representing 19.7% of respondents, failed the final question and were excluded from the dataset.

## F ADDITIONAL DETAILS ON THE STATISTICAL ANALYSIS

### F.1 INPUT OF LINEAR REGRESSION

To support the statistical analysis presented in §3, we employed linear regression models. Each model was trained to predict a single outcome metric (fairness, efficiency, Alice's self-gain, or Bob's self-gain) within a specific family. The input features included all game-defining parameters (as listed in Table 1), along with identifiers for the two players involved in the game.

To represent player identity, we introduced two additional features, each indicating one of the players.[20] Within each game family, we combined the parameters defining the market (T, CI and MA/MT - see Table 1) into a single parameter named *market*, representing the interaction between the three original parameters. We do so because their interaction defines the structure of the market itself, and the total number of combinations was small enough to allow reliable modeling. Thus, for example, in bargaining games, the *market* parameter became a vector with eight entries, corresponding to each possible combination arising from the Cartesian product of these three market-defining parameters.

Each parameter was represented using a one-hot encoding vector, where each possible value of the parameter was assigned a distinct entry. The entry corresponding to the actual value of the parameter was set to 1, while all other entries were set to 0. For instance, the parameter $M$ was split into three distinct parameters: $M = 10^2$, $M = 10^4$ and $M = 10^6$, each taking the value of 1 if the respective condition held and 0 otherwise. Figure 20 illustrates this feature representation.

### F.2 BASELINE MODELS

Table 8 presents the default parameter values used in our statistical analysis (see §3). These values serve as the reference points for estimating the impact of each parameter value on the target metrics. The selection of default values is based on the simplest model in game theory.

---

[20]When analyzing model interactions (e.g., in Tables 21, 22, 23), we replaced these two features with a single feature representing the pairwise identity of the two LLMs.

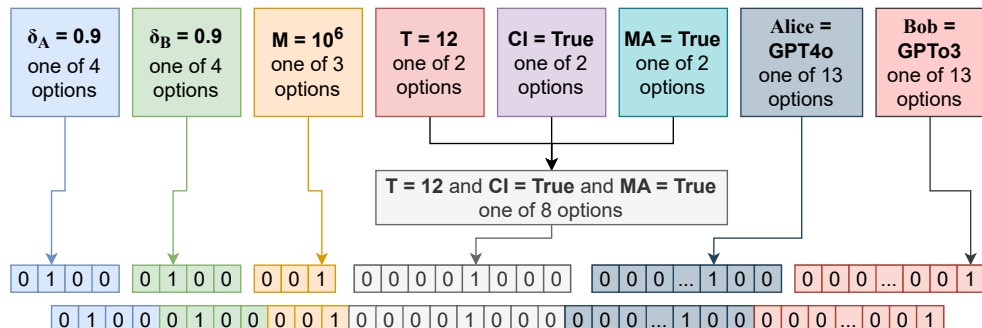

Figure 20: Illustration of feature encoding for linear regression in a bargaining game. Each row corresponds to a single bargaining game instance, encoded as a binary feature vector. The input includes game-defining parameters ($\delta_A$, $\delta_B$, $M$), alongside a composite *market* feature derived from the interaction of three specific parameters: $T$, $CI$, and $MA$. This *market* feature is one-hot encoded with eight possible values, representing all combinations of the three. Player identities (Alice and Bob) are encoded using separate one-hot vectors with 13 possible values each. The binary vectors at the bottom of the figure illustrate how all components are concatenated into a single input row, where each 1 indicates the active value of a categorical parameter for this particular bargaining game instance.

Table 8: Default parameter values used in the statistical analysis. $T = \infty$ means a very large value of $T$, unknown to the players. CI = Complete Information. MA = Textual messages allowed.

| Bargaining | | Negotiation | | Persuasion | |
|---|---|---|---|---|---|
| $\delta_A$ | 0.9 | $F_A$ | 1 | p | 0.5 |
| $\delta_B$ | 0.9 | $F_B$ | 1 | v | 1.25 |
| $M$ | $10^4$ | $M$ | $10^4$ | M | $10^4$ |
| $T$ | $\infty$ | $T$ | 1 | $T$ | 20 |
| CI | True | CI | True | CI | True |
| MA | False | MA | False | Messages type | Binary |
| | | | | Buyer type | Long-living |

## F.3 PREDICTIVE VALIDITY OF LINEAR REGRESSION

Section 4 describes the method used for our analyses, based on beta coefficient interpretation in linear regression. In this appendix, we demonstrate that linear regression achieves strong predictive performance compared to state-of-the-art (SOTA) models on tabular regression tasks.

For each of the three game families (bargaining, negotiation, persuasion) and each of the four evaluation metrics (Efficiency, Fairness, Alice's self-gain, Bob's self-gain), we trained XGBoost (Chen & Guestrin (2016)) and CatBoost (Prokhorenkova et al. (2018)) as SOTA baselines.

The dataset was randomly split into 80% training and 20% testing. All models received the full feature set as input (see §4 for details) and were tasked with predicting the target metric.

Table 9 summarizes the results. We find that the linear model performs on par with, and occasionally outperforms, the more complex SOTA models, with only minor differences in RMSE. These findings suggest that the relationships between game parameters and outcomes are largely linear, and thus support the use of beta coefficients as a reliable and interpretable tool for analyzing how input features, such as game configurations and agent identities, influence performance metrics.

Table 9: Root Mean Squared Error (RMSE) for each model across families and metrics. Values are reported as mean ± standard deviation over 100 random seeds.

| Family | Metric | Linear Regression | XGBoost | CatBoost |
|---|---|---|---|---|
| **Bargaining** | Alice Self Gain | 0.134 ± 0.001 | 0.134 ± 0.001 | 0.134 ± 0.001 |
| | Bob Self Gain | 0.129 ± 0.002 | 0.129 ± 0.002 | 0.129 ± 0.002 |
| | Efficiency | 0.131 ± 0.002 | 0.131 ± 0.002 | 0.131 ± 0.002 |
| | Fairness | 0.157 ± 0.003 | 0.156 ± 0.003 | 0.156 ± 0.003 |
| **Negotiation** | Alice Self Gain | 0.137 ± 0.014 | 0.137 ± 0.014 | 0.136 ± 0.014 |
| | Bob Self Gain | 0.139 ± 0.013 | 0.138 ± 0.012 | 0.138 ± 0.012 |
| | Efficiency | 0.333 ± 0.004 | 0.331 ± 0.004 | 0.331 ± 0.004 |
| | Fairness | 0.090 ± 0.003 | 0.088 ± 0.003 | 0.088 ± 0.003 |
| **Persuasion** | Alice Self Gain | 0.356 ± 0.003 | 0.353 ± 0.003 | 0.352 ± 0.003 |
| | Bob Self Gain | 0.834 ± 0.011 | 0.842 ± 0.011 | 0.840 ± 0.011 |
| | Efficiency | 0.388 ± 0.004 | 0.380 ± 0.004 | 0.378 ± 0.004 |
| | Fairness | 0.379 ± 0.004 | 0.372 ± 0.005 | 0.371 ± 0.004 |

## G  ADDITIONAL RESULTS

### G.1  LLMs PERFORMANCE (Q2)

In this appendix, we present additional result graphs that complement those shown in §4.

In this sub-appendix, we present graphs that help address Q2: How do different LLMs behave in strategic interactions, and which models achieve fair, efficient, and high self-gain outcomes?

Figures 21, 22, and 23 show how the pair of models playing bargaining, negotiation, and persuasion games influenced the efficiency and fairness of the game.

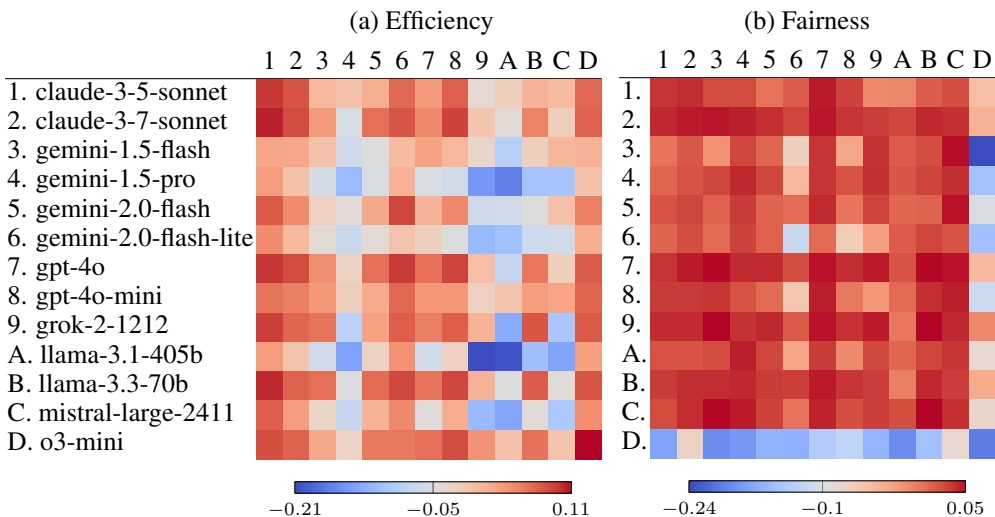

Figure 21: The effect of the identities of the two players (rows: Alice, columns: Bob) on efficiency and fairness in Bargaining games, reported relative to the mean outcome.

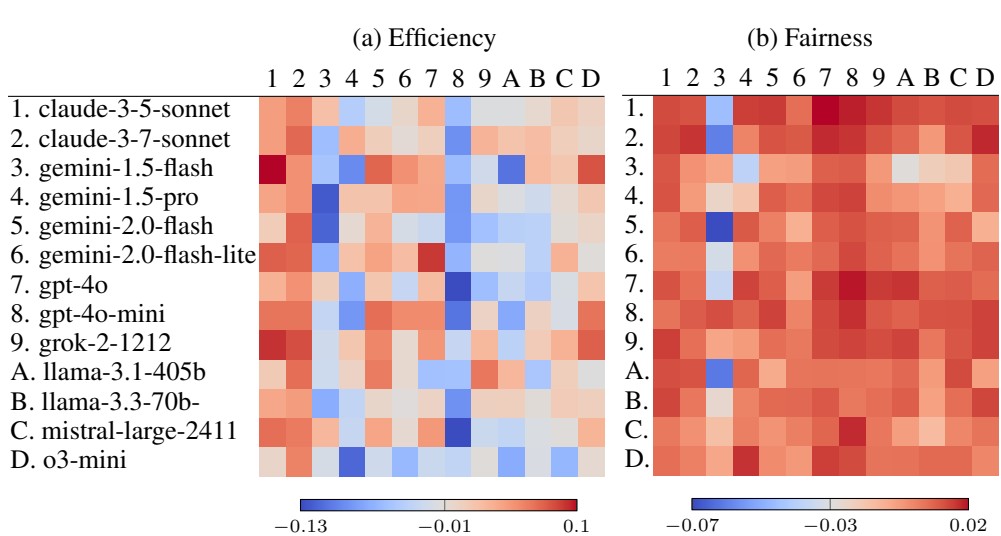

Figure 22: The effect of the identities of the two players (rows: Alice, columns: Bob) on efficiency and fairness in Negotiation games, reported relative to the mean outcome.

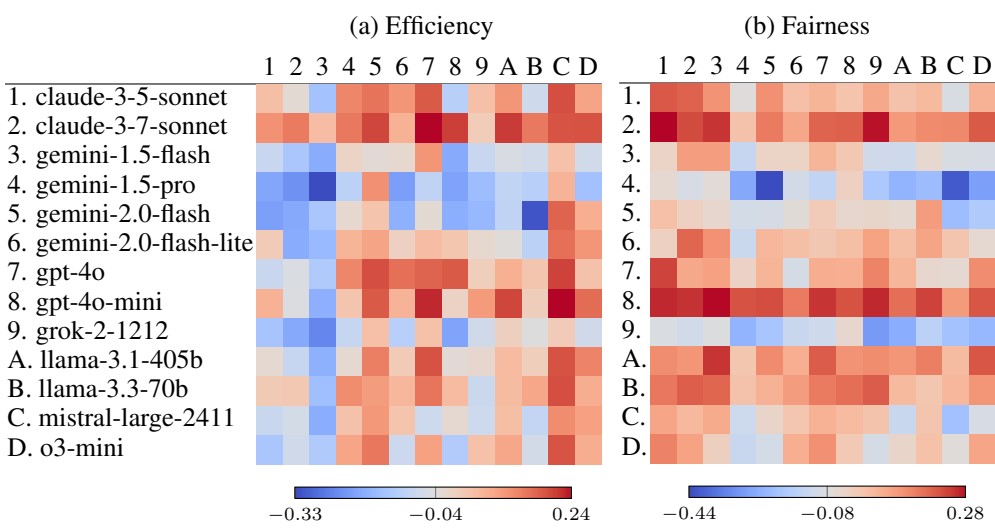

Figure 23: The effect of the identities of the two players (rows: Alice, columns: Bob) on efficiency and fairness in Persuasion games, reported relative to the mean outcome.

