# OpenReview forum: "GLEE: A Unified Framework and Benchmark for Language-based Economic Environments"
_ICLR.cc/2026/Conference — ICLR 2026 Conference Withdrawn Submission_

### Official Review · Reviewer_2JVm · 2025-10-26

**Soundness:** 2
**Presentation:** 3
**Contribution:** 2
**Rating:** 4
**Confidence:** 4

**Summary:**

The authors propose a economic benchmark comprised of negotiation, persuasion, and bargaining instances. The authors claim that despite these realms being separately studied in prior works, the benchmark provides a standardization over the three games allowing for comparability and generalization of findings. The authors conducted experiments on the proposed benchmark, showing insights related to fairness and efficiency, LLM performance, and comparison with human performance.

**Strengths:**

Overall I think the presentation of the paper is good, the authors develop some interesting observations in Section 3, and the benchmark itself is well-presented. The experimental data also seem sufficient.

**Weaknesses:**

I think the paper faces notable headwind in persuading on significance and novelty, particular as it lays out honestly the vast number of literature beginnning at least 2 years ago studying LLM on each of the three topics they investigate: bargaining, negotiation, and persuasion. Some of the works, as authors mentioned in line 85, include Abdelnabi et al. (2023), Bianchi et al. (2024), .... (I'll omit a list of relevant work that's easily accessible with keyword search). The authors argue that prior "approaches have varied widely across different studies" and propose a framework for modeling more aligned with theoretical works in economics. Arguably, none of the modeling decisions or aspects here are completely unexplored (or without analog) in past work, but I do see the rationale for standardization.

However, the case that standardization itself leads to new economic insights that could not be obtained by separately running prior, task-specific benchmarks is underdeveloped. The paper emphasizes heterogeneity across prior studies and the need for comparability, but to meet the acceptance bar it should more directly demonstrate discoveries that *require* a unified setup rather than reproduce patterns that could plausibly be found per-task., otherwise practitioners can simply choose to use separate benchmarks on each subtopic, arriving at similar conclusions and insights. This seems especially true since while the framework shares labels for "efficiency" and "fairness", the three families instantiate these over different primitives-discounted split of a divisible pie (bargaining), price-based welfare and mid-valuation proximity (negotiation), and state-aligned buy/no-buy behavior (persuasion)-so inter-game comparisons largely reflect differing economic models and metric semantics, not a single, common notion. For example, the paper reports interesting cross-family patterns (e.g., message allowance improves bargaining metrics but degrades both efficiency and fairness in persuasion), yet one concern might be that such qualitative patterns could be recovered by taking three best-in-class, task-specific benchmarks and running the same models.

Other minor points: The manuscript highlights an anchoring effect in bargaining (e.g., correlation 0.63 when humans are Bob and LLMs set the first offer, vs. 0.18 when roles flip). This is intriguing but I think given the space constraint, this seem more like a possible hypothesis and less convincing - fundamentally correlational given the current design and alternative explanations (selection, unobserved heterogeneity, wording effects) remain open. This might be an interesting work in itself but demand a more rigorous treatment.

Lastly, I understand how fast things move in the LLM landscape, so I’m not expecting the authors to rerun everything, but it’s worth noting that the human experiments are conducted against Gemini-1.5-Flash, which does not invalidate the findings but is a legacy model.

**Questions:**

- Can the authors better articulate the gains from standardization, and perhaps offer anything concrete (perhaps, show an example that cannot be recovered by running three separate, task-specific benchmarks).

- Beyond notational alignment, is there a formal bridge that makes notions like efficiency and fairness comparable across the different primitives? For example, can you specify a shared welfare objective and show that each family’s metric is a monotone transform of it under stated assumptions? If not, does it breaks comparability?

- Do the authors foresee materially different results with more capable reasoning models (e.g., better planning/tool use)?

---

### Official Review · Reviewer_s61x · 2025-10-27

**Soundness:** 2
**Presentation:** 2
**Contribution:** 2
**Rating:** 2
**Confidence:** 2

**Summary:**

The authors propose a unified benchmark framework, GLEE, designed to evaluate the behavior of LLMs in language-driven economic game environments. GLEE systematically constructs three core economic scenarios and defines a multidimensional parameter space on this basis, establishing standardized evaluation metrics such as efficiency, fairness, and self-gain.

**Strengths:**

1. Based on classical economic models, the paper introduces a unified benchmark framework for language-based economic environments.
2. The experiments are extensive, covering 13 LLMs, 1,320 game configurations, and 80,000 interactions.

**Weaknesses:**

1. Although the experiments reveal the complex interaction between language and rational behavior, the paper lacks a deep explanation of the underlying economic mechanisms.
2. The paper reads more like a social-science-style experimental report, and it is unclear whether it fully aligns with ICLR’s expectations.
3. The writing could be improved; readers without an economics background may find it difficult to follow. Some illustrative figures could help, and the excessive use of footnotes seems inconsistent with typical writing styles in AI research papers.

**Questions:**

1. Have you considered extending GLEE to multi-agent cooperation settings?
2. In Table 3, why are all the results for gemini-1.5-flash equal to zero?
3. Does the linguistic style used by LLMs in persuasion or bargaining affect the experimental outcomes? How do you control for the impact of LLM output variability on simulation results?

---

### Official Review · Reviewer_BTqc · 2025-11-01

**Soundness:** 3
**Presentation:** 3
**Contribution:** 2
**Rating:** 2
**Confidence:** 2

**Summary:**

This paper presents GLEE, a framework for evaluating the behavior of Large Language Models (LLMs) in language-based economic games. The goal of GLEE is to provide a comparative tool for assessing the performance of LLMs in various economic scenarios and enable their comparison to human players. The authors define three game families (bargaining, negotiation, and persuasion), collect data from 587K decisions across 80K games with 13 LLMs, and analyze how different market parameters affect efficiency, fairness, and agent performance.

**Strengths:**

1. The authors introduce a clear parametrization of a large, general, and representative set of sequential, two-player, language-based games. This parametrization is inspired by both the relevant economic literature, as well as advances in the LLM agents literature. The three game families are well-motivated from classical economic models (Rubinstein bargaining, bilateral trade, cheap talk games).

2. The scale of data collection is large: including954K games between LLMs and from 3,405 games involving human players. Testing 13 different LLMs across 1,320 configurations provides substantial coverage.

**Weaknesses:**

1. The restriction to two-player sequential games, limits real-world applicability. Many economic interactions involve multiple parties or simultaneous decision-making.

2. The specific values chosen for game parameters (discount factors, valuations, time horizons) appear not well justified. The paper lacks sensitivity analysis or principled justification for these choices. For instance, why these specific discount factor values (0.8, 0.9, 0.95, 1) ?

3. The paper uses fixed prompts across all models and configurations. Given known sensitivity of LLMs to prompting, this choice may confound model-specific effects with prompt compatibility. No analysis of prompt robustness is provided.

**Questions:**

Please see problems in weaknesses

---

### Official Review · Reviewer_4N1V · 2025-11-01

**Soundness:** 2
**Presentation:** 2
**Contribution:** 3
**Rating:** 2
**Confidence:** 3

**Summary:**

This paper introduces GLEE, a unified framework for evaluating Large Language Models (LLMs) in language-based economic games. The authors formalize three fundamental game families: bargaining, negotiation and persuasion.

They develop an open-source platform and collect a massive dataset of 587K decisions across 80K games involving 13 LLMs, plus 3,405 human-vs-LLM interactions.

Using linear regression analysis, the study reveals that (1) market parameters significantly impact efficiency and fairness, with complex interactions between complete information and linguistic communication; (2) no single LLM dominates across all metrics, with performance heavily dependent on opponent choice.

**Strengths:**

1. Principled framework with strong theoretical grounding: The paper bridges AI/NLP and economic theory by formalizing games inspired by foundational models. The consistent parameterization across game families and unified metrics enable systematic comparison—a major improvement over prior ad-hoc approaches.

2. Large-scale empirical contribution: 587K decisions across 1,320 configurations with comprehensive human data collection (rigorous attention checks, 26.8% exclusion rate).

**Weaknesses:**

Critical: Limited Technical Contribution for ICLR
1. No algorithmic innovation: The paper presents a data collection framework and statistical analysis but proposes no new methods, architectures, or training techniques. Linear regression (Section 3) is standard; XGBoost/CatBoost comparison (Table 9) only validates existing methods.
2. This is more suited to domain-specific venues (e.g., ACL for NLP applications, EC/WINE for computational economics) or dataset/benchmark tracks rather than ICLR's main research track focused on learning algorithms and representations.
3. No learning methods proposed: Unlike ML benchmarks that drive algorithmic innovation (e.g., ImageNet → ConvNets, GLUE → pre-training methods), GLEE provides no clear direction for how to improve LLM economic reasoning. Missing:
    * Training objectives optimizing for efficiency/fairness.
    * Reinforcement learning methods for economic games.

Methodological Limitations

1. Shallow mechanistic understanding:
    * Why does complete information reduce efficiency in bargaining with messages? No analysis of actual message content or causal mechanisms.
    * Findings remain descriptive without actionable insights for model improvement.
2. Limited scope:
    * No prompt/temperature robustness testing despite Appendix D.2 showing prompts vary significantly.
    * Single LLM (Gemini-1.5-flash) for all human experiments—unclear if findings generalize.

**Questions:**

1. Algorithmic direction: Can you propose concrete methods GLEE could evaluate? E.g., RLHF variants optimizing efficiency+fairness, multi-agent training protocols, or prompting strategies?
2. Robustness: What is sensitivity to prompt variations, temperature, and opponent model choice? Without this, practical applicability is unclear.
3. Multiple testing: Should Table 2 confidence intervals use Bonferroni/FDR correction given 12 simultaneous tests?
4. Venue fit: What algorithmic innovations does GLEE enable that justify ICLR submission?

**Details Of Ethics Concerns:**

1. IRB approval exists (may exist but not documented in paper).
2. Human data release follows appropriate privacy protocols.

---

### Note · Authors · 2025-11-19

I have read and agree with the venue's withdrawal policy on behalf of myself and my co-authors.